# Coronary artery Segmentation in Cardiac CT Angiography Using 3D Multi-Channel U-net

**Yo-Chuan Chen[1], Yi-Chen Lin[1], Ching-Ping Wang[4], Chia-Yen Lee[4], Tzung-Dau Wang[2],Wen-Jeng Lee[3],Chung-Ming Chen[1*]**

[1]*Institute of Biomedical Engineering, National Taiwan University*

[2]*Cardiovascular Center and Division of Cardiology, Department of Internal Medicine, National Taiwan University Hospital, Taiwan*

[3]*Department of Medical Imaging, National Taiwan University Hospital, Taiwan*

[4]*Department of Electrical Engineering, National United University*

**Editors:** Under Review for MIDL 2019

## Abstract

Vessel stenosis is a major risk factor in cardiovascular diseases (CVD). To analyze the degree of vessel stenosis for supporting the treatment management, extraction of coronary artery area from Computed Tomographic Angiography (CTA) is regarded as a key procedure. However, manual segmentation by cardiologists may be a time-consuming task, and present a significant inter-observer variation. Although various computer-aided approaches have been developed to support segmentation of coronary arteries in CTA, the results remain unreliable due to complex attenuation appearance of plaques, which are the cause of the stenosis. To overcome the difficulties caused by attenuation ambiguity, in this paper, a 3D multi-channel U-Net architecture is proposed for fully automatic 3D coronary artery reconstruction from CTA. Other than using the original CTA image, the main idea of the proposed approach is to incorporate the vesselness map into the input of the U-Net, which serves as the reinforcing information to highlight the tubular structure of coronary arteries. The experimental results show that the proposed approach could achieve a Dice Similarity Coefficient (DSC) of 0.8 in comparison to around 0.6 attained by previous CNN approaches.

**Keyword:** coronary artery, vessel stenosis, segmentation, multi-channel, U-net

## 1. Introduction

Cardiovascular disease accounts for 45% of non-communicable diseases in 2015 [1]. Vessel stenosis in coronary artery is considered to be the major risk in CVD. Narrowing of the lumen limits the blood flow and affects the oxygen supply to cardiomyocytes, leading to myocardial infarction.

Computed Tomography angiography (CTA) images is one of the widely used noninvasive imaging modalities in coronary artery diagnosis due to its superior image resolution. CTA images with contrast agents can make coronary arteries more visible. To analyze the degree of vessel stenosis for supporting the treatment management, extraction of coronary arteries from Computed Tomographic Angiography (CTA) is regarded as a key procedure. However, manual delineation of coronary arteries is a time-consuming task, and presents a significant inter-observer variation. Clinically, high-quality extraction of coronary arteries is essential for stenosis quantification.

Various conventional image segmentation algorithms have been proposed previously to reconstruct 3D cardiovascular structures, such as region-based methods [2], edge-based methods [3][4], tracking-base methods [5],

learning-based methods [6][7], and so on. Since 2016, the applications of deep learning in medical images have grown considerably [8]. Several deep learning approaches based on CNN have been proposed to reconstruct coronary arteries in CTA. Nevertheless, these deep learning approaches show relatively limited performance, e.g., 0.5975 and 0.66 in terms of Dice Similarity Coefficient (DSC) [10] [11]. One possible reason is the size of the coronary artery is relatively small in comparison with the surrounding tissues. Another possible reason is it is challenging to distinguish the coronary arteries from other tubular structures, e.g., the coronary veins [12].

Although great efforts have been made for automatic reconstruction of coronary arteries in CTA, the results remain far from satisfactory due to the potential intensity ambiguity between coronary arteries and surrounding tissues or plaques. To overcome the difficulties caused by intensity ambiguity, in this paper, a 3D multi-channel U-Net [13] architecture is proposed to automatically reconstruct the coronary arteries in CTA.

## 2. Methods

In addition to the original CTA image, the main idea of the proposed approach is to incorporate the vesselness map into the input of the U-Net, which serves as the reinforcing information to highlight the tubular structure of coronary arteries. Figure 1 shows the proposed network architecture. In the proposed architecture, the input is composed of two channels of the same volume of interest (VOI) (32*32*32), one from the original CTA image and the other from the vesselness map derived by applying Frangi filtering [15] to the original CTA image. The segmentation result is given in the output (32*32*32) of the U-net. Rectified Linear Unit (ReLU) is used as the activation function during convolution and deconvolution, and max pooling is used for downsampling. The network loss function is based on the concept of the similarity between the output image and the ground-truth image. Thus, Dice Similarity Coefficient is chosen as loss function for the network.

The data used in this study include 33472 training samples (11 cases), 5683 and 12223 validation samples(2 cases), 6841 and 7028 testing samples(2 cases). Each case is subject to a vascular enhancement filter, to obtain candidate regions. Irrelevant tissues in the thoracic images, for example, bone tissue and lung tissue, are removed by using bounding box, thresholding, and morphological processing. The vessel region of the training set is established by skeletonizing the image of the candidate region, and the skeletonized points are used as the center for the VOIs to generate the VOIs of the vessel region. In addition, data augmentation is realized by flipping and rotating the VOIs of the vessel region. On the other hand, since the background area is large, data augmentation is adopted for the background region, and VOIs are randomly selected from the background region such that the number of samples of vessel region and background region is 1:1. The testing set VOI is generated from the entire cardiac region.

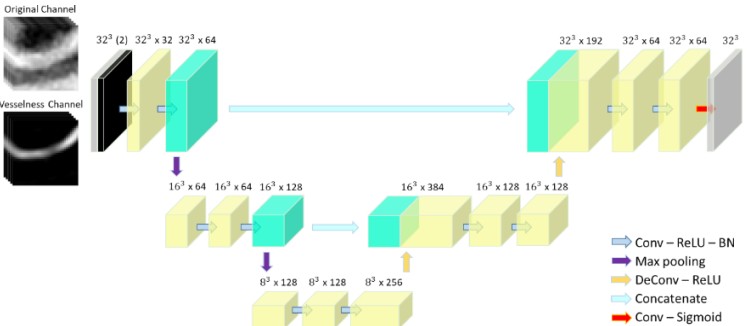

Figure 1. Deep learning network, the numbers above are feature map size in the format (#$size^3$x #channel)

| Case | DSC-3D | | |
|---|---|---|---|
| | Mean | Max | Max + LCC |
| Validation 1 | 0.8003 | 0.7528 | 0.8080 |
| Validation 2 | 0.7998 | 0.6834 | 0.7727 |
| Test 1 | 0.8039 | 0.7284 | 0.7964 |
| Test 2 | 0.7691 | 0.6810 | 0.7639 |
| Test 3 | 0.7853 | 0.7323 | 0.7972 |
| Test 4 | 0.7916 | 0.7465 | 0.8426 |
| Test 5 | 0.7790 | 0.7017 | 0.8150 |
| Test 6 | 0.7865 | 0.7277 | 0.7980 |
| Test 7 | 0.8213 | 0.7706 | 0.8364 |
| Test 8 | 0.7951 | 0.6811 | 0.8301 |
| Average | 0.7932 | 0.7206 | **0.8060** |

Table 1. performance of each postprocessing in Validation cases and Test cases

## 3. Results

The output result of the network VOI can be reconstructed accordingly by corresponding the original coordinates of the VOI centers, and overlapping the reconstruction processed by taking the maximum value or the mean value of each overlapping pixel. We then take the largest connected component(LCC) twice to exclude the fragmented non-coronary artery artifacts and keep the left and right coronary artery. The maximum processing is used in the reconstruction stage to find the coronary artery structure as much as possible to retain the entire coronary artery and remove non-coronary artery regions. The reconstruction result is shown in Figure 2.

From Table 1, although the performance of taking the maximum value is lower than taking the mean value, it can be improved by applying the largest connected component achieving 0.8 Dice Similarity Coefficient performance.

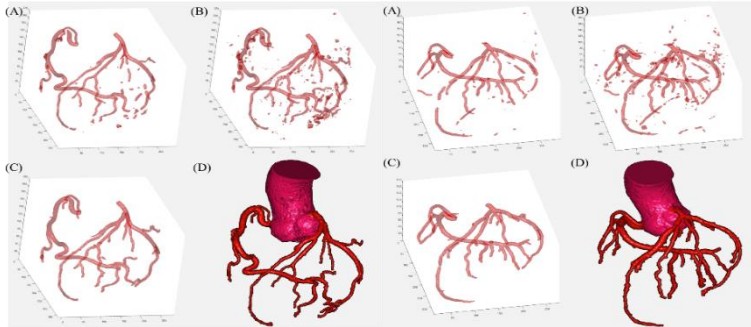

Figure 2. the reconstruction result of each kind of postprocessing, part (A) is processed by taking mean, part(B) is processing by taking maximum, part (C) is processing by taking maximum and largest connected component, part (D) is manually annotation result by commercial software.

## 4. Discussion and Conclusion

Other approaches in deep learning for coronary artery segmentation show relatively limited performance[10] [11]. In comparison, the proposed approach achieves 0.8 DSC performance. The reason might be due to U-net having the same output size as the input image, therefore it is much more suitable than normal CNN in segmentation task. Secondly, the multi-channel concept, allowed the model to have tubular structural information during training, aiding the network to learn more effectively. In Kirişli et al. [14], ground-truth is the average region of the three observers, so there would be three performances for each observer. It is worth noting that the best observer has only 0.79

performance in DSC, thus the algorithm has shown to outperform the observer's performance.

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
