# OpenReview forum: "Coronary artery Segmentation in Cardiac CT Angiography  Using 3D Multi-Channel U-net"
_MIDL.io/2019/Conference/Abstract — MIDL Abstract 2019_

### Official Review · AnonReviewer2 · 2019-04-27
**Interesting work to incorporate the vesselness map**

**Rating:** 3
**Confidence:** 3

**Review:**

It is interesting work to incorporate the vesselness map to the segmentation network. The results also look promising. However, it is not clear how much gain we get by adding the additional vesselness map on the same dataset. It would be worth exploring for future work.

---

### Official Review · AnonReviewer1 · 2019-05-01
**yet another proof (in a new application): incorporating prior can improve segmentation**

**Rating:** 3
**Confidence:** 2

**Review:**

Authors used a 3D multi-channel U-Net architecture  for automatic 3D coronary artery reconstruction from CTA. The main contribution relies on incorporating vesselness map into the architecture for final segmentation. The problem can be considered like other vision and imaging problems where additional information (prior) enhances the segmentation when there is valuable information in the prior or it constraints the segmentation procedure to a smaller region where convergence is easier.

The main concern of the paper is that the vesselness map is based on mostly manual / semi-automated procedure of extracting vessel information, and providing this information into the U-net architecture. If vesselness map includes wrong information, or fail due to some pathological formation, then the system will not be reliable (or at least we readers do not know the uncertainty that it can cause).

The use of multi-channel U-Net is not new. There are different names that scientists are willing to give to their architectures although essentially many of them are doing the same thing. Here the basic architecture is accepting secondary input from vessel ness map, but its contribution is implicit (results are showing some improvement though). A better fusion strategy can be done to prove the complementary information.

Despite major issues, the application can be considered incrementally new as I did not see exact same way of artery segmentation from CTA (editors can correct me if I am wrong).

Minor:
- there are unclear details about the test and. validation, and also not clear how patches are combined, and performance of the network in a clear and neat manner (some texts can be clearly cut, table can be put inside the text instead of having large scape,  and more results can be put as this is fairly an application paper, requiring more evidence and comprehensive results)
-figure 2 is not publication quality. Try to use open-source tools such as ITKSNap or others to make smooth reconstruction so that it can be understood what is going on. The surfaces rendered in figure 2 do not have clinical/qualitative value for detailed inspection.
-grammar and flow have some problems.

---

### Decision · Program_Chairs · 2019-05-06
**Acceptance Decision**

Accept